# Chronic wounds in Sierra Leone: Searching for Buruli ulcer, a NTD caused by *Mycobacterium ulcerans*, at Masanga Hospital

Helen R. Please[1]*, Jonathan H. Vas Nunes[2,3,4], Rashida Patel[1], Gerd Pluschke[5,6], Mohamed Tholley[3], Marie-Thérésè Ruf[5,6], William Bolton[1], Julian A. Scott[1], Martin P. Grobusch[2,3,4], Håkon A. Bolkan[3,4,7], Julia M. Brown[8], David G. Jayne[1]

1 Leeds Institute of Medical Research, University of Leeds, Leeds, United Kingdom, 2 Center of Tropical Medicine and Travel Medicine, Department of Infectious Diseases, University of Amsterdam, Amsterdam, The Netherlands, 3 Masanga Medical Research Unit, Masanga, Sierra Leone, 4 CapaCare International, Trondheim, Norway, 5 Swiss Tropical and Public Health Institute, Basel, Switzerland, 6 University of Basel, Basel, Switzerland, 7 Department of Clinical and Molecular Medicine, Norwegian University of Science and Technology, Trondheim, Norway, 8 Institute of Clinical Trials Research, University of Leeds, Leeds, United Kingdom

* helenplease@gmail.com

**Data Availability Statement:** All relevant data are within the manuscript and its Supporting Information files.

## Abstract

### Background

Chronic wounds pose a significant healthcare burden in low- and middle-income countries. Buruli ulcer (BU), caused by *Mycobacterium ulcerans* infection, causes wounds with high morbidity and financial burden. Although highly endemic in West and Central Africa, the presence of BU in Sierra Leone is not well described. This study aimed to confirm or exclude BU in suspected cases of chronic wounds presenting to Masanga Hospital, Sierra Leone.

### Methodology

Demographics, baseline clinical data, and quality of life scores were collected from patients with wounds suspected to be BU. Wound tissue samples were acquired and transported to the Swiss Tropical and Public Health Institute, Switzerland, for analysis to detect *Mycobacterium ulcerans* using qPCR, microscopic smear examination, and histopathology, as per World Health Organization (WHO) recommendations.

### Findings

Twenty-one participants with wounds suspected to be BU were enrolled over 4-weeks (Feb-March 2019). Participants were predominantly young working males (62% male, 38% female, mean 35yrs, 90% employed in an occupation or as a student) with large, single, ulcerating wounds (mean diameter 9.4cm, 86% single wound) exclusively of the lower limbs (60% foot, 40% lower leg) present for a mean 15 months. The majority reported frequent exposure to water outdoors (76%). Self-reports of over-the-counter antibiotic use prior to presentation was high (81%), as was history of trauma (38%) and surgical interventions prior to enrolment (48%). Regarding laboratory investigation, all samples were negative for

**Funding:** This research was funded by the National Institute for Health Research (NIHR), using UK aid from the UK Government to support global health research. 16/137/44, NIHR Global Health Research Group on Surgical Technologies, DGJ, JS, and JMB. https://www.nihr.ac.uk/explore-nihr/funding-programmes/global-health.htm The views expressed in this publication are those of the author(s) and not necessarily those of the NIHR or the UK Department of Health and Social Care. The funders had no role in study design, data collection and analysis, decision to publish, or preparation of the manuscript.

**Competing interests:** The authors have declared that no competing interests exist.

BU by microscopy, histopathology, and qPCR. Histopathology analysis revealed heavy bacterial load in many of the samples. The study had excellent participant recruitment, however follow-up proved difficult.

## Conclusions

BU was not confirmed as a cause of chronic ulceration in our cohort of suspected cases, as judged by laboratory analysis according to WHO standards. This does not exclude the presence of BU in the region, and the definitive cause of these treatment-resistance chronic wounds is uncertain.

### Author summary

Chronic wounds constitute a significant surgical burden to low- and middle-income countries; however, their aetiology often remains poorly understood. This study improves our understanding of wound aetiology through tissue analysis of chronic leg wounds suspected to be caused by Buruli ulcer (BU). BU is a neglected tropical disease caused by infection with *Mycobacterium ulcerans*, and remains severely under-researched. There is a lack of testing facilities in regions surrounding endemic countries which makes prevalence difficult to determine, with a particular paucity of data from Sierra Leone (SL). This study identified twenty-one patients with wounds suspected to be caused by BU who presented to Masanga Hospital (Tonkonili District, Sierra Leone) between February and March 2019. Tissue samples were acquired from the wounds and transported to a European tropical health laboratory for analysis. Significant bacterial loads were demonstrated in the samples. However, the gold-standard molecular tests recommended by World Health Organisation (WHO) revealed no cases of BU. These results suggest that BU is not a major cause of chronic wounds in the Northern Province of Sierra Leone. Our conclusions cannot necessarily be generalised to other regions of Sierra Leone, therefore further studies in other geographical districts are required.

## Introduction

Lack of access to surgical care is a public health crisis in low- and middle-income countries (LMICs), as highlighted by The Lancet Commission on Global Surgery [1]. In 2015, 5 billion people worldwide lacked access to surgical care, with a deficit of 143 million operations annually, and 25% of surgery recipients facing catastrophic financial consequences. Sierra Leone is a sobering example with <10% of its surgical need currently being met, and non-specialists performing the majority of operations [2]. Further research into the barriers and solutions to surgical care in such settings is therefore imperative [3–4]. This study focused on chronic wound aetiology, the management of which has been identified as a global priority [5]. Wounds represent a large burden of disease in Sierra Leone. They are often stigmatizing and highly disabling, and their management is labour and resource intensive.

Buruli ulcer (BU) is a severe cutaneous disease caused by infection with *Mycobacterium ulcerans (M. ulcerans)* which produces mycolactone, a highly potent cytotoxin responsible for the chronic necrosis and ulceration of BU. It is primarily endemic in West and Central Africa [6], however, there is limited data on the presence of BU in Sierra Leone, with only 28 cases

confirmed in 2011 and a single case in 2008, according to the World Health Organisation (WHO) Global Health Observatory Data Repository at time of publishing [7]. There are known cases in the countries surrounding Sierra Leone, with a reported 549 cases in Guinea, and 353 cases in Liberia between 2011 and 2017 [8].

Diagnosis of BU can be difficult. Polymerase chain reaction (PCR) and histopathology demonstrate the best sensitivity (85–89% and 90%, respectively) and positivity rates (70–90% and 90% respectively), as summarised by Portaels et al. [9]. However, direct smear examination (DSE) is more widely available, and as such PCR is often reserved for DSE-negative suspected cases. New point-of-care rapid diagnostic tests with higher sensitivity are currently being developed with the aim of improving testing availability [10].

Wounds with unknown aetiology often require surgical treatments such as debridement and skin grafting, whereas BU is treatable with anti-mycobacterial therapy and appropriate wound dressing. However, if left undetected BU can lead to large, destructive, ulcerative lesions; with 25% of cases detected too late to prevent disability [11]. The transmission of BU is poorly understood, although there is a strong epidemiological association with stagnant water. Early detection and treatment reduce morbidity and potentially crippling treatment costs. The WHO Global Buruli Initiative started in 1998 incorporating research, policy, and control measures to reduce the burden of disease in areas known to be endemic [12]. The aim of this study was to confirm or exclude BU in this group of patients presenting to Masanga Hospital with chronic wounds, through laboratory analysis of wound tissue samples according to WHO standards.

## Methods

### Ethics statement

The study was reviewed by the Masanga Medical Research Unit's Scientific Review Committee and subsequently received ethics approval from the University of Leeds (MREC17-110) and the National Ethics Committee of Sierra Leone (February 2019). All patients approached gave their consent for participation in the study. Written consent was obtained, with thumbprint signatures used in cases of illiteracy, an established method at Masanga Hospital. Specific consent was obtained to transfer tissue samples (swabs and biopsies) to the SwissTPH for analysis.

This single-centre, observational, cross-sectional study was carried out at Masanga Hospital, a medium-sized district hospital in the Northern Province of Sierra Leone widely reputed for its surgical focus and surgical training programme delivered by the non-governmental surgical organisation CapaCare [13]. Its growing wound care service attracts patients nationwide (>300 inpatients annually) to provide general wound care, burn management, debridement, skin grafts, and amputation. The majority of cases are treated empirically due to a lack of sophisticated diagnostic facilities. Clinical suspicion is that some of the chronic wound burden is caused by BU. The hospital's growing research capacity provided the opportunity to carry out a study to diagnose BU in suspicious wounds, and collect necessary demographic and clinical data for patients with chronic wounds.

The study design comprised of tissue sample acquisition and wound swab collection or fine needle aspiration in suspected BU cases and transportation of these samples to a specialist laboratory, the Swiss Tropical and Public Health Institute (SwissTPH), in Basel, Switzerland, for tissue analysis, real-time PCR and direct smear examination to confirm/exclude the diagnosis of BU (Fig 1). The study was reviewed by the Masanga Medical Research Unit's Scientific Review Committee and subsequently received ethics approval from the University of Leeds (MREC17-110) and the National Ethics Committee of Sierra Leone (February 2019).

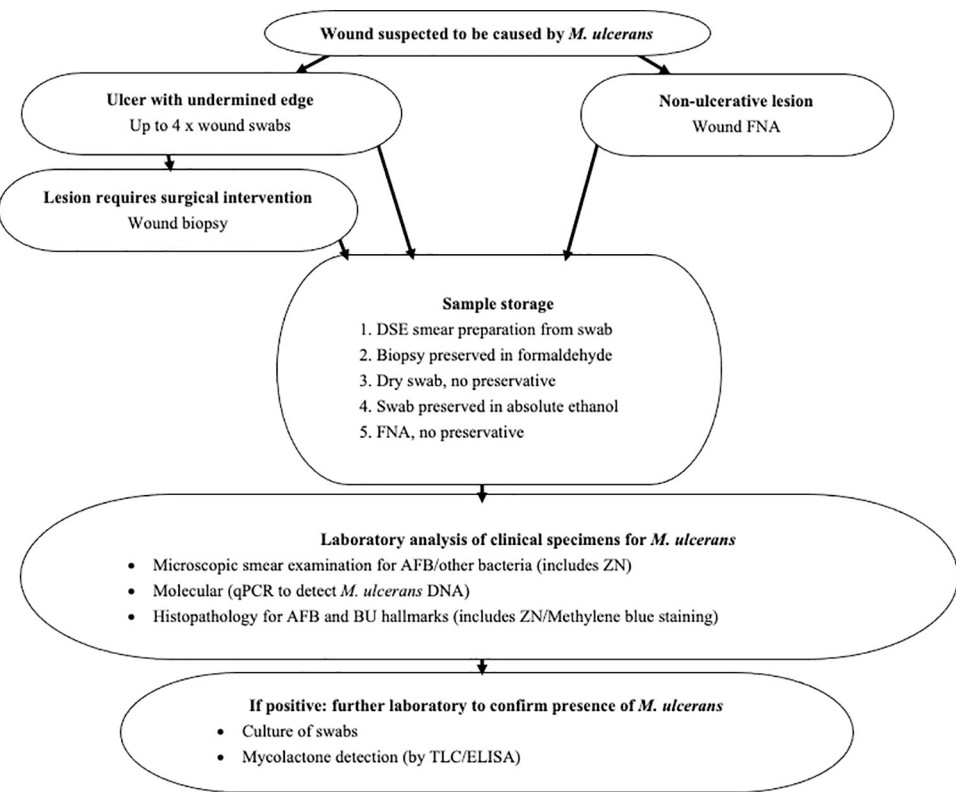

**Fig 1. Study Protocol.** Process for suspected BU cases of tissue sample acquisition, storage by multiple methods, transportation and analysis.

The main purpose of this study was to improve our understanding of the aetiology of chronic skin ulcers, specifically by confirming or excluding BU in chronic wounds locally identified as suspicious of BU. The variables of interest in the participant database included recruitment rates, participant demographics, risk exposure, and wound characteristics such as size, depth, appearance. The outcome measures for the BU sampling included results of qPCR testing, histopathological analyses, and direct microscopic examination of wound smears after Ziehl-Neelsen (ZN) staining.

## Identification of participants

Eligible participants with wounds suspected to be caused by BU were identified from inpatient and outpatient settings over a 4-week period (Feb—March 2019). Participants were provided with information sheets, translated where necessary. Written consent was obtained, with thumbprint signatures used in cases of illiteracy, an established method at Masanga Hospital. Specific consent was obtained to transfer tissue samples (swabs and biopsies) to the SwissTPH for analysis. There was no change in wound management resulting from patients being included or not included in the study. All participants received regular treatment for their wounds as per local protocol. For the majority this included regular wound dressing, and in some cases surgical intervention such as debridement.

For inclusion in the study, participants had to have an ulcer that was clinically suspected by the local healthcare staff to be caused by *M. ulcerans* infection. Adults and children were eligible provided they could provide informed consent, and were otherwise well (American Society of Anaesthesiologists, ASA, grade ≤3 [14]). Suspicion of BU was based on several criteria to

encompass any lesions exhibiting one of the classical four forms of BU, including: painless itchy nodules, plaques, small ulcerations, and extensive ulcerations. In particular, painless ulcers were examined for the presence of undermined edges; chronic non-healing wounds (>1 month duration), and participants identified as having risk factors associated with BU (including living in rural areas, spending long periods of time in stagnant waters/rivers/wet-beds/flooded rice fields). Although both ulcerating and non-ulcerating lesions were included in the study design, no patients with non-ulcerating lesions suspicious of BU were identified in the study period. Exclusion criteria included chronic wounds where an alternative aetiology for the wound was apparent, in particular exclusion of any wounds suspected to be caused by diabetes mellitus, arterial disease, venous disease, or a specified NTD such as leprosy.

Healthcare staff at the hospital were given training on BU, including clinical recognition, epidemiology, treatment options. This was delivered in a lecture form with a question-and-answer session, and discussion time. Twelve healthcare staff from the hospital attended, including doctors, nurses and other healthcare workers working in the wound dressing department, surgical department, and on the general wards.

Baseline clinical data and demographics were collected from participants via history, examination, and medical notes. These were recorded on paper forms under unique study identifiers, which were later entered into a secure participant database. The clinical data collected included medical history, social history, drug history, surgical history, vital signs, laboratory results, suspected diagnosis, and anatomical wound location (Tables 1 and 2). As part of the medical history, all participants were asked about current and past medical problems using both general questions with open answer, as well as structured questions relating to the aetiology of lesion formation. This included enquiring if there was a current or previous history of diabetes, leprosy, or infective diagnosis; see note in Table 2 for the full list. Contact details were taken in order to facilitate follow-up, planned at one- and three-months following tissue acquisition.

## Quality of life score

A wound-specific quality of life (QoL) score was calculated using the validated Wound QoL (Questionnaire on quality of life with chronic wounds) tool, which measures the disease-specific, health-related QoL of patients with chronic wounds [15]. The one-page questionnaire was completed by participants, comprising seventeen questions which are self-assessed in retrospect to the preceding seven days. The questions consider wound-related impairments, including: pain, smell, discharge, effect on sleep, treatment burden, happiness, frustration, anxiety, fear of further wounds, fear of injury to wound, limit to mobility, stair-climbing, effect on activities of daily living, limits to leisure activities, limits to socialising, dependency on others, and financial burden. For each question, participants select one of five possible answers coded with numbers i.e. 'Q1. In the last seven days my wound hurt. . .' 0 = not at all, 1 = a little, 2 = moderately, 3 = quite a lot or 4 = very much. A Wound-QoL global score was calculated by averaging all question response values, with a maximum global score of four and minimum of zero. The global score could only be computed if at least 75% of the items had been answered. Results were used to measure changes in wound-specific QoL over time.

## Collection of clinical specimens

All tissue samples were collected and analysed using standard procedures as recommended by the WHO for laboratory confirmation of BU cases [9]. In the case of ulcerated lesions, multiple swab specimens were taken from the lesion by passing the swab around the undermined edge rather than the centre of the lesion, to ensure that the areas with the highest load of bacteria

**Table 1. Demographic & wound characteristics of participants.**

| | Age (yrs) | Gender | Weight (kg) | Occupation | Highest education level | Often in water outside? (river/ rice field) | Number of wounds | Location of largest wound | Largest wound diameter (cm) | Months since wound started | Limb oedema | Wound depth | Undermined edge |
|---|---|---|---|---|---|---|---|---|---|---|---|---|---|
| | DEMOGRAPHICS | | | | | | WOUND CHARACTERISTICS | | | | | | |
| 1 | 15 | M | 50–60 | Student | High school | Yes | Single | Left lower leg | 2.2 | 12 | Yes | Skin | Yes |
| 2 | 18 | F | 10–20 | Student | High School | Yes | Single | Left foot | 12.6 | 3 | Yes | Skin | Yes |
| 3 | 19 | M | 60–70 | Student | High school | Yes | Multiple (Bilateral) | Left lower leg | 9 | 12 | No | Skin | Yes |
| 4 | 23 | M | 50–60 | Trader | High school | Yes | Multiple (Unilateral) | Right foot | 9 | 7 | No | Skin | Yes |
| 5 | 25 | F | 90–100 | Student | High school | Yes | Single | Right lower leg | 8 | 6 | No | Skin | Yes |
| 6 | 25 | F | 60–70 | Trader | High school | Yes | Single | Left foot | 6 | 12 | No | Skin | Yes |
| 7 | 25 | M | 50–60 | Mechanic | Illiterate | Yes | Single | Right foot | 10 | 24 | No | Skin | Yes |
| 8 | 27 | F | 50–60 | Student | High school | Yes | Single | Left foot | 8.5 | 4 | No | Skin | Yes |
| 9 | 27 | M | 60–70 | Office | High school | Yes | Single | Right foot | 11 | 1 | No | Skin | Yes |
| 10 | 29 | M | 70–80 | Farmer | High school | No | Single | Left lower leg | 10 | 12 | No | Skin | Yes |
| 11 | 32 | M | 50–60 | Trader | Illiterate | Yes | Single | Right lower leg | 15 | 5 | No | Bone | Yes |
| 12 | 37 | M | 60–70 | Trader | High school | Yes | Single | Left foot | 14 | 11 | No | Skin | Yes |
| 13 | 38 | M | 70–80 | Office | University | Yes | Single | Right foot | 20 | 12 | Yes | Tendon | Yes |
| 14 | 42 | M | 70–80 | Farmer | Illiterate | Yes | Single | Right foot | 14 | 12 | No | Skin | Yes |
| 15 | 42 | M | 70–80 | Farmer | Illiterate | No | Single | Left foot | 5 | 8 | No | Skin | Yes |
| 16 | 42 | F | 80–90 | Trader | High school | Yes | Single | Left lower leg | 6.5 | 4 | Yes | Skin | Yes |
| 17 | 43 | M | 80–90 | Other | Illiterate | Yes | Single | Left lower leg | 19.7 | 20 | Yes | Tendon | Yes |
| 18 | 52 | F | 50–60 | Farmer | Illiterate | No | Single | Left foot | 3 | 12 | No | Skin | Yes |
| 19 | 56 | F | 80–90 | Unemployed | High school | Yes | Single | Left lower leg | 16.3 | 34 | No | Skin | Yes |
| 20 | 63 | M | 80–90 | Tailor | Illiterate | No | Multiple (Bilateral) | Left lower leg | 2.5 | 80 | No | Skin | Yes |
| 21 | 65 | F | 60–70 | Fishing | Illiterate | No | Single | Right foot | 10.6 | 24 | Yes | Skin | Yes |

Participant demographics as reported by the participant. Wound characteristics as reported by medical staff (expectation of 'months since wound started', reported by the participant). Participant numbers are unrelated to Study ID to maintain confidentiality; Yrs: years old; kg: kilograms; Bilateral: wounds on left & right (i.e. both legs); Unilateral: wound(s) confined to left or right

were sampled, as per WHO protocol regarding swab sampling for BU [9]. Swab samples were stored in 8 mL vials; one dry, one immersed in ethanol covering the entire tip of the swab, and one in culture medium. One swab was also used to prepare a DSE by spreading the sample from the swab tip onto a clean microscopy slide. This was only performed when there was a sufficient sample for each modality; otherwise dry and ethanol swabs were prioritised. For participants undergoing active surgical treatment, such as wound debridement, biopsies were carried out by the local surgeon. Biopsy was performed under local anaesthesia and involved a small wedge biopsy (1–2 cm wide, 0.5 cm thick). As mycobacteria are distributed heterogeneously in lesions, two biopsies were taken from each ulcer. Where possible, one biopsy was

**Table 2. General health of participants & types of tissue samples collected.**

| Participant | PAST MEDICAL HISTORY | | GENERAL HEALTH | | | | | TISSUE SAMPLES COLLECTED | | | | |
|---|---|---|---|---|---|---|---|---|---|---|---|---|
| | Medical history* (not necessarily concurrent)* | Surgical history | Haemoglobin at presentation | General Tiredness* | Palpable Lymph Nodes | Weight Loss* | Quality of life score (max 4) | DSE | Biopsy | Dry Swab | Culture swab | Ethanol Swab |
| 1 | Malaria | Nil | 9.4 | Yes | No | Yes | 2.5 | No | No | Yes | Yes | Yes |
| 2 | Malaria | Nil | 12 | Yes | No | Yes | 3.2 | Yes | Yes | Yes | Yes | Yes |
| 3 | Nil | SSG | 9.8 | Yes | No | No | 1.5 | Yes | No | Yes | Yes | Yes |
| 4 | Malaria | SSG | 11.1 | No | No | Yes | 2.5 | Yes | No | Yes | Yes | Yes |
| 5 | Malaria | Nil | 10.7 | Yes | No | No | 1.9 | Yes | No | Yes | Yes | Yes |
| 6 | Meningitis & Malaria | Nil | 11.7 | No | No | No | 2.0 | No | Yes | Yes | Yes | Yes |
| 7 | Nil | SSG & Debrid. | 11.4 | Yes | No | No | 1.1 | Yes | No | Yes | Yes | Yes |
| 8 | Sickle cell disease | Amputation | 10.1 | No | No | Yes | 2.8 | Yes | No | Yes | Yes | Yes |
| 9 | Malaria | Nil | 10.4 | No | No | Yes | 2.9 | Yes | Yes | Yes | Yes | Yes |
| 10 | Yellow fever & Malaria | SSG | 11.3 | Yes | No | Yes | 2.8 | Yes | Yes | Yes | Yes | Yes |
| 11 | Malaria | Amputation | 6.6 | Yes | No | Yes | 2.5 | Yes | No | Yes | Yes | Yes |
| 12 | Nil | SSG | 9.4 | Yes | No | No | 2.9 | Yes | No | Yes | Yes | Yes |
| 13 | Hydrocoelectomy | SSG & Debrid. | 13.4 | No | No | Yes | 3.6 | Yes | Yes | Yes | Yes | Yes |
| 14 | Malaria | Nil | 10.2 | No | Yes | No | 2.2 | Yes | No | Yes | Yes | Yes |
| 15 | Measles & Yellow fever & Malaria | Nil | 7.2 | Yes | Yes | Yes | 2.0 | Yes | No | Yes | Yes | Yes |
| 16 | Malaria | Nil | 11.4 | No | Not recorded | Not recorded | Not recorded | Yes | Yes | Yes | Yes | Yes |
| 17 | Yellow fever & Malaria | Nil | 11.8 | Yes | No | Yes | 3.8 | Yes | Yes | Yes | Yes | Yes |
| 18 | Yellow fever & Malaria | SSG | 8.5 | Yes | No | Yes | 1.1 | Yes | No | Yes | Yes | Yes |
| 19 | Yellow fever & Malaria | Nil | 12.4 | No | No | No | 3.8 | No | Yes | Yes | Yes | Yes |
| 20 | Malaria | SSG | 8.8 | Yes | No | No | 1.9 | No | No | Yes | Yes | Yes |
| 21 | Yellow fever & Malaria & Measles | Nil | 12.7 | Yes | No | Yes | 2.0 | Yes | Yes | Yes | Yes | Yes |

Past medical history and general health of participants, as well as types of tissue samples collected. DSE: direct smear examination, SSG: split skin graft, Debrid.: Debridement.

*self-reported by participants (other aspects recorded by medical staff).

**Unless stated, negative for current/past history of: diabetes, pyoderma, leprosy, onchocerciasis, kidney disease, stroke, anaemia, heart failure, HIV, asthma/ emphysema, tuberculosis, scabies, arthritis, malaria, measles, yellow fever, Lassa fever, pneumonia, schistosomiasis, dengue fever, cholera & meningitis

taken from near the centre of the wound, and another closer to the wound edge. Where multiple ulcers were present, the largest wound was biopsied. Biopsy samples were fixed with 10% neutral-buffered formalin. The biopsy technique for non-ulcerative lesions included a FNA sample taken with a needle and stored in 8mL vials with no preservative. However, this technique was not used as no patients with non-ulcerating lesions suspicious of BU presented to Masanga Hospital during the study period.

All tissue samples were stored at 4˚C and transported to the laboratory in Switzerland (SwissTPH) with a temperature logger to ensure temperature stability. Transportation logistics were assisted by the Foundation for Innovative New Diagnostics (FIND) [16].

A total of five different types of tissue collection and storage were used to optimise sample types for the individual analyses (Fig 1). The initial three methods allow detection of *Mycobacterium* spp. including: DSE, which is a standard method to microscopically identify acid-fast bacilli (AFBs); tissue biopsies, which are suitable to detect both AFBs and the characteristic histopathological hallmarks of BU; and dry swabs, which are suitable to detect *M. ulcerans* DNA by PCR. In addition, two further types of swab samples were taken, which would be analysed only if the initial tests were positive. These included: ethanol swabs, which are optimal to keep mycolactone intact for the detection of this macrolide toxin by thin layer chromatography (TLC) or enzyme-linked immunosorbent assays (ELISA); and culture swabs, which are optimal to protect the viability of *M. ulcerans* cultivation. As the latter two methods are very labour-intensive, they would be performed only if one of the three initial tests were positive. The transportation logistics from Sierra Leone to Switzerland, resulting time from tissue acquisition to analysis, and concern of participant loss to follow-up made it unfeasible to carry out the various sample techniques at different timepoints. Therefore, all five methods of sample collection (DSE, tissue biopsy, dry swab, ethanol swab, and culture swab) were performed in the one-month study timeframe.

### Laboratory analysis of clinical specimens

Sample analysis at SwissTPH included microscopic smear examination, qPCR testing, and histopathological analyses [17] for the presence of *M. ulcerans*; the protocol is summarised in Fig 1.

For direct microscopic detection of acid-fast bacilli (AFB), wound exudate smears on glass slides were fixed by pulling the glass slide three times through a flame. Slides were stained with Ziehl-Neelsen (ZN, detection of mycobacteria) and Methylene blue (staining of nucleic acid) [9] and embedded into Eukitt (Sigma-Aldrich, St. Louis, United States) mounting medium. Entire slides were analysed for the presence of AFB with a Leica DM2500 microscope using the 40x and the 100x objectives. For qPCR analysis, DNA was extracted from the swab samples (or FNA if performed), and qPCR was performed as described by Lavender & Fyfe [18], using the real-time PCR primers IS2404TF with the sequence AAAGCACCACGCAGCATCT, IS2404TR with the sequence AGCGACCCCAGTGGATTG, and the minor-groove binding probe IS2404TP with the sequence 6FAM-CGTCCAACGCGATC-MGBNFQ. Spiked controls were included to ensure that no PCR inhibitors were present in the extracts. The SwissTPH laboratory in which this analysis was carried out achieved 100% concordance in the external quality assessment program (EQAP) for the PCR-based detection of *M. ulcerans* in clinical samples [19].

For histopathological examination, fixed tissue specimens were dehydrated, embedded into paraffin, and cut into 5μm thin sections. After deparaffinisation and rehydration, the sections were stained with ZN/Methylene blue (Sigma-Aldrich) according to WHO standard protocols [9]. The sections were then mounted on glass slides, coverslipped, and viewed under a Leica DM2500 microscope at 40x and the 100x magnification.

## Results

All patients approached gave their consent for participation in the study. In the 4-week period of recruitment, twenty-one participants were identified by local doctors and nursing staff as having potential BU and consented to tissue sampling.

### Demographics of participants with wounds suspected to be caused by BU

Of the twenty-one participants with wounds suspicious of BU, there were a higher number of males (62%), and the mean age was 35 years old (range 15 to 52 years). Participants came from

a large geographical region around Masanga Hospital; and had a range of employments, including: farming, office work, mechanic, trader, student, tailor, fisherman, and unemployment. Over half of participants had an education level to high school qualification (55%), one had a bachelor's degree (4.8%), and the remainder were illiterate (38%). A high proportion of participants reported often spending time in water in an outdoor environment (76%), whether through farming or recreational activities, see Table 1.

In terms of co-morbidities and alternative aetiologies for the wounds, no participants reported a current or previous history of diabetes, and none of the participants were diagnosed with diabetes based on a blood glucose test. No participants had a history of other aetiologies for chronic wound infection, including vascular disease. The most common self-reported past medical condition was malaria (71% of participants). No participants had verified concomitant morbidities at time of presentation with their wounds. In terms of current symptoms, over half the participants described tiredness (62%) and weight loss (57%) at the time of presentation, although there was no suspicion of malignancy on clinical assessment. There was an erroneous lack of recording for 3 pieces of information for one participant (participant 16, Table 2, 'not recorded'). The fact that these items were missed was only identified after the participants had been discharged and were uncontactable. For completeness, this participant was included in the study because all other inclusion criteria and data points were met. One patient was initially considered to be included in the study but was later excluded due to a past medical history of treated leprosy. Swabs were taken in this case, and returned negative for BU, however the decision was made to exclude the patient as their history of leprosy was considered a potential confounder, particularly as their wound had been present for 36 months.

The average global QoL score was 2.3 (range 1.1 to 3.8), out of a maximum global QoL score of 4. It was not possible to record serial QoL scores due to the high rate of loss to follow-up, therefore changes in wound-specific QoL over time could not be evaluated. Despite best attempts to contact participants for follow-up, there was a frequent change in phone number or participants were only contactable through a relative. Many participants were unable to provide a specific address, only the name of their village or town.

## Clinical features of wounds suspected to be caused by BU

The majority of participants had single wounds (86%); two participants had multiple wounds bilaterally (i.e. on each leg), and one participant had multiple wounds unilaterally. All wounds were on the anterior aspect of the lower limb, either of the feet (60%) or the lower leg (40%). No upper limb chronic wounds were observed. All wounds suspicious of BU were large ulcerating lesions with a mean largest diameter of 9.4cm (range 2.2 to 20cm). No participants presented with painless nodules or plaques. The majority of wounds were scored as skin deep ulceration (86%); two wounds had tendons exposed; one had bone exposed. The wounds had been present for a mean of 15 months (range 1 to 80 months).

All participants enrolled as BU-suspicious cases had ulcerating lesions with undermined edges, therefore swabs were taken in every case. No participants with non-ulcerating lesions suspicious of being caused by BU presented during the study period, therefore FNAs were not required. Depending on the amount of tissue acquired, the swab tissue was stored in as many of the storage methods as possible (17 DSE preparations, 10 tissue biopsies in formaldehyde, 21 dry swabs, 21 ethanol preserved swabs, no FNAs, see Table 2).

Regarding clinical history, 52% of participants reported the lesion starting as a boil or swelling, 38% reported some association with trauma, and no participants reported development from an insect bite. The majority of participants reported using over-the-counter antibiotics

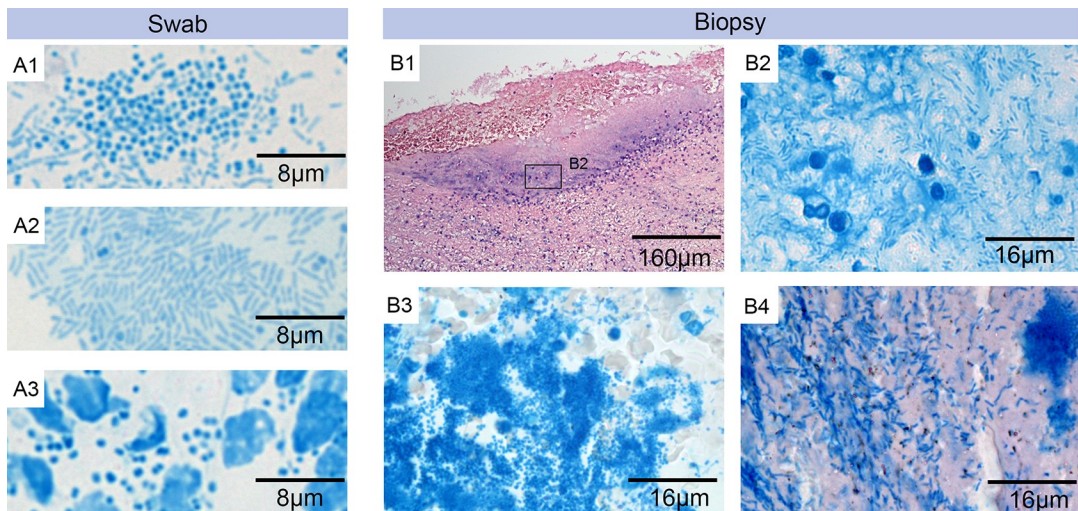

**Fig 2. Laboratory analysis.** Microscopic examination of swab smears (A1 –A3) and histological sections (B1 –B4) of biopsies from wounds suspected to be BU lesions. Specimens were either stained with ZN-Methylene blue (A1-A3 and B2-B4) to detect AFBs (pink) and nucleic acid, implicating DNA from secondary infections (blue); or with Haematoxylin-Eosin (B1). A1-A3: Staining of wound exudate from different swabs revealed a heavy infection with rods and/or cocci but no AFBs. A1: rods and cocci (participant 14). A2: rods (participant 11). A3: cocci and leukocytes (participant 4). B1: overview of excised tissue specimen revealing a secondary infection at the open ulcer surface and the underlying tissue (participant 21). B2: higher magnification of this area revealed the presence of rods but no AFBs (participant 21), B3: the same higher magnification at a different area revealed a strong infection with cocci but no AFBs (participant 21) B4: staining of tissue sections from a different participant revealed the presence of large numbers of rods but again no AFBs (participant 6).

prior to presentation (81%, predominantly ampicillin, metronidazole, ceftriaxone and doxycycline either alone or in combination); the length or number of treatment courses was difficult to ascertain. Ten out of the twenty-one participants had undergone surgical interventions, either during their hospital stay, or previously. Two had split skin grafts and debridement, six had split skin grafts only, and two had amputations. The indication for these surgical interventions were dependant on the individual cases, according to local hospital management protocols. See Table 2 for further details.

## Laboratory studies for BU diagnosis

Sample collection and transportation from Sierra Leone to Switzerland was successful, with a validated temperature logger demonstrating a consistent temperature of the samples throughout the journey. No AFBs were found when the ZN-Methylene blue stained DSEs were examined (Fig 2). However, Methylene blue staining revealed a significant burden of other bacteria including cocci and rods (Fig 2 A1-A3). Molecular analysis with qPCR from swab samples was negative for *M. ulcerans* DNA in all samples. Spiked controls demonstrated that no PCR inhibitors were present in the extracts. In addition, the histopathology analyses of the biopsy tissue specimens revealed no AFBs in ZN-Methylene blue staining, but did show heavy infections with a broad range of other microorganisms (Fig 2 B1-B4). On the basis of these analyses, including negative qPCR, microscopic smear examination, and histopathology for *M. ulcerans*, as per WHO recommendations, BU was excluded in all cases. As these analyses on the DSE, biopsy, and dry swab results were negative for *M. ulcerans*, there was no indication for carrying out TLC or ELISA analysis for detection of mycolactone in the ethanol and culture swabs. It was not possible to ascribe a definitive pathogenic aetiology to individual cases.

## Discussion

The study demonstrated that it is possible to obtain tissue samples from participants with lesions suspicious of BU in Sierra Leone, and securely transport samples to a specialist reference laboratory in Europe. Study recruitment was excellent, with all participants approached giving consent. No *M. ulcerans* was diagnosed in any of the twenty-one suspected cases when analysed using state-of-the-art diagnostic tools recommended by WHO. The consistency of the transportation temperature log demonstrates that it is feasible to move samples from a region lacking sophisticated analysis to a distant centralised laboratory. Due to the significant time investment and logistics of transportation, and in order to maximise efficiency of resources, the samples were stored to allow batch transportation and analysis. As a result, the final laboratory results were available five months following samples acquisition. This delay is unlikely to have influenced the results, however it does highlight the need for point-of-care rapid diagnostic/lateral flow tests for M. ulcerans, for which work is ongoing [10].

The heavy bacterial presence in the tissue samples, including cocci and rods, is not unusual in chronic wounds from tropical countries, and has been reported also in BU lesions [20]. These findings may reflect wound colonization, which is not necessarily harmful, however heavy colonization can interfere with wound healing. Poor initial wound management, particularly in traumatic wounds, can lead to heavy infection and progression to chronic lesions which increase in size over time. In such cases, systemic antibiotics can be considered, taking care not to encourage antibiotic resistance. Long-term hospitalisation can lead to multi-resistant bacterial colonisation, with some arguing this is a case for de-centralization of wound management [21]. It was not possible to link the presence of non-BU bacteria to the individual ulcer aetiology.

A high proportion of participants reported frequent exposure to water in outdoor environments, a well-known risk factor for BU demonstrated through studies in similar geographical regions including Cote d'Ivoire, Ghana, and Togo [6]. Due to the wide breadth of aetiologies for ulcerative lesions, the study design attempted to exclude wounds that were likely caused by an alternative aetiology, such as diabetes or leprosy. This was achieved by a combination of clinical judgement, the local investigations available, and self-reporting of participants' past medical histories. There were no confounding reports of alternative wound aetiology. Incidental note was made of high rates of malaria and yellow fever, which do not cause ulcerating skin lesions.

The design of this study purposefully included both adults and children in order to include the population in West Africa in which BU presents most frequently (those under 15 years of age). However, no paediatric patients presented to Masanga hospital with wounds. It may be that adults are more likely to seek, or be able to access, healthcare. A future study might address this by recruiting patients from a community, rather than hospital, setting.

We found that patients with chronic wounds presenting to Masanga Hospital were mainly young men, with wounds entirely sited on the lower limbs. Classically, BU lesions are located in >90% cases on the limbs, with about two-thirds on the lower and one-third on the upper limbs, and less frequently the trunk, and sites on the neck/head [6]. Around half were employed and/or educated to high school qualification, but there may have been some bias, with those who have the funds and knowledge to travel to a hospital being over-represented. Most wounds were single, on the lower limb/ankle/foot, of skin depth, large size (mean diameter 9.4cm), and had been present for just over one year. Surprisingly, given the size and duration of the ulcers, QoL scores were moderate (average global score of 2.3 out of total of 4 for maximum impairment), which might suggest a tolerance to what is perceived to be a common condition.

Many participants reported a traumatic incident potentially starting the process (38%), however the histories were vague. It may be that such wounds start with minor trauma and poor initial wound management in the community leads to secondary bacterial colonisation/ infection. Wound infection with *Staphylococcus aureus*, *Pseudomonas aeruginosa*, and beta-hemolytic streptococci may delay wound healing when bacterial loads exceed $10^6$ colony forming units/g [22]. The histological examination and wound smear analyses suggests that such organisms may play a central role in the chronicity of the wounds. Methylene blue DNA staining was adequate to demonstrate that cocci and rods were present in a significant amount, however no specific staining or culture was performed to determine bacterial species present. For future studies, bacterial cultures at the site of investigation should be performed, as done in similar research [20].

Although BU can be diagnosed based on the detection of the infecting organism, clinical evaluation also plays an important role, with several studies suggesting initiation of treatment with BU-specific antibiotics based primarily on clinical suspicion in line with WHO clinical and epidemiological criteria [23,24]. The detection of AFB is challenging due to the low yield on conventional AFB stains. The AFB positivity with ZN staining is quoted at 0–44% in various studies [25,26], and there is a lack of standardisation on identifying AFBs on smears [27]. In terms of the standard crosslinking fixation with formalin needed to analyse histological specimens, some studies suggest that AFB detection is underestimated [28]. Nonetheless, this remains the gold-standard as novel staining methods with high sensitivity are yet to be developed. PCR remains the most accurate test known for *M. ulcerans*.

Chronic wounds are a significant burden on limited healthcare resources in LMICs, as illustrated by half of the study participants undergoing some form of surgical treatment. A greater understanding of the prevalence, aetiology, and treatment of chronic wounds is required and might be aided by the development of chronic wound registries, allowing assimilation of epidemiological and clinical information. In addition, there is scope for integration of wound care, triaging patients with wounds to determine who are most suitable to be treated in the community or in hospital, and a need for more community-based wound care through education and peripheral clinics.

A limitation of the study was that it was not possible to collect follow-up data. The reasons for this are complex, with an almost 100% loss of participants to follow-up as soon as they left hospital. Contact details for each participant were collected at the time of study recruitment, however many of the participants were uncontactable, or could only provide contact details of relatives. Participants often lived a long distance from the hospital, they were non-contactable by phone, they had no fixed address, and there was no postal service for contact. Such barriers to research are widespread in LMICs. Our findings therefore represent a snapshot of chronic lower limb ulceration, rather than a longitudinal analysis.

Another limitation of the study is that it is an observational cross-sectional study, performed on a subset of a people with chronic wounds presenting to a single study site. Therefore, whilst it is important to determine the prevalence of BU in Sierra Leone, the study is limited in scope to a single hospital with a low number of study participants. Therefore our conclusions cannot be generalised to other regions of Sierra Leone. The presence of previously identified cases in Sierra Leone [7] strongly suggests BU may still be present in the country. This is supported by studies which use statistical modelling generated from the geographical locations of previously confirmed cases, enabling suitability mapping of the regions in which BU is likely to occur. These models suggest that BU is likely to be present beyond the range of known endemic areas, specifically including Sierra Leone. The study demonstrates that Sierra Leone has suitable conditions for the disease to be present; however, what is not clear is whether additional factors are required for transmission to humans. This may be a reason that

the presence of BU can be transient, and geographically focal. Therefore, further work is required to define the geographical regions of Sierra Leone in which BU is most likely to occur, as this is largely unknown.

A further short-coming is that although both ulcers and non-ulcerating lesions suspicious of BU were included, no participants with non-ulcerating lesions (and therefore requiring FNA rather than swabs) were identified. This may be due to patients with non-ulcerating lesions being less likely to seek medical attention. This limitation might be overcome by utilising a community-based sampling approach. In addition, although diabetes was not suspected in any of the participants, most participants did not give a clear history of other aetiologies, such as leprosy, diabetes, or vascular disease.

In conclusion, we have been unable to provide definitive proof of BU in highly suspicious cases of chronic ulceration in a district general hospital in Sierra Leone, despite applying established state-of-the-art diagnostic tools. This does not exclude the presence of BU in the country, and the cause of many chronic wounds remains uncertain. We have shown that it is possible to undertake diagnostic studies in resource-poor environments, with potential to draw on expertise at a geographically distant location if local laboratory expertise and resources are limited. A greater research effort is now needed, with improved longitudinal evaluation, in order to understand the aetiology of chronic ulcers so that effective treatment strategies can be instigated to alleviate the suffering and reduce the burden on healthcare resources.

## Supporting information

**S1 STROBE Checklist. Checklist of items that should be included in reports of cross-sectional studies, completed for this study.**
(DOCX)

## Author Contributions

**Conceptualization:** Helen R. Please, Jonathan H. Vas Nunes, David G. Jayne.

**Data curation:** Helen R. Please, Jonathan H. Vas Nunes, Rashida Patel, Mohamed Tholley.

**Formal analysis:** Helen R. Please, Gerd Pluschke, Marie-Therésè Ruf.

**Funding acquisition:** David G. Jayne.

**Investigation:** Helen R. Please, Jonathan H. Vas Nunes.

**Methodology:** Helen R. Please, Gerd Pluschke.

**Project administration:** Helen R. Please, Jonathan H. Vas Nunes, William Bolton, Julian A. Scott, Martin P. Grobusch, Håkon A. Bolkan, Julia M. Brown, David G. Jayne.

**Supervision:** David G. Jayne.

**Writing – original draft:** Helen R. Please.

**Writing – review & editing:** Jonathan H. Vas Nunes, Gerd Pluschke, Martin P. Grobusch, Håkon A. Bolkan, David G. Jayne.

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
