## [Decision Letter · Decision Letter 0]

23 May 2021

Dear Dr. Please,

Thank you very much for submitting your manuscript "Chronic wounds in Sierra Leone: searching for Buruli ulcer, a NTD caused by Mycobacterium ulcerans." for consideration at PLOS Neglected Tropical Diseases. As with all papers reviewed by the journal, your manuscript was reviewed by members of the editorial board and by several independent reviewers. In light of the reviews (below this email), we would like to invite the resubmission of a significantly-revised version that takes into account the reviewers' comments. 

Your manuscript submission has been reviewed by two experts in the field. While it is important to determine the prevalence of Buruli ulcer in Sierra Leone, the reviewers agree that this study was very limited in scope. This should be reflected in the title and the limitations of the data should be explicitly indicated in the Introduction and Discussion. There are a number of methodological issues to be addressed as well as the age range of the patients. Please address all the issues raised by both reviewers in your reply and indicate these in your revised submission.

We cannot make any decision about publication until we have seen the revised manuscript and your response to the reviewers' comments. Your revised manuscript is also likely to be sent to reviewers for further evaluation.

Sincerely,

Paul J. Converse

Associate Editor

Mathieu Picardeau

Deputy Editor

Dear Dr. Please,

Your manuscript submission has been reviewed by two experts in the field. While it is important to determine the prevalence of Buruli ulcer in Sierra Leone, the reviewers agree that this study was very limited in scope. This should be reflected in the title and the limitations of the data should be explicitly indicated in the Introduction and Discussion. There are a number of methodological issues to be addressed as well as the age range of the patients. Please address all the issues raised by both reviewers in your reply and indicate these in your revised submission.

Reviewer's Responses to Questions

**Key Review Criteria Required for Acceptance?**

**Methods**

-Are the objectives of the study clearly articulated with a clear testable hypothesis stated?

-Is the study design appropriate to address the stated objectives?

-Is the population clearly described and appropriate for the hypothesis being tested?

-Is the sample size sufficient to ensure adequate power to address the hypothesis being tested?

-Were correct statistical analysis used to support conclusions?

-Are there concerns about ethical or regulatory requirements being met?

Reviewer #1: (No Response)

Reviewer #2: This is an observational crosssectional study, performed on a subset of a peoples with chronic wound received in a single study site, namely the Masanga Hospital. Therefore, even if the study design and the method are well described, I'll suggest, given the method of recruitment of participants, and the very low number of study participant, to reconsider the title of the manuscript as followed: Chronic wounds in Sierra Leone: searching for Buruli ulcer, a NTD caused by Mycobacterium ulcerans, at the Masanga Hospital.

**Results**

-Does the analysis presented match the analysis plan?

-Are the results clearly and completely presented?

-Are the figures (Tables, Images) of sufficient quality for clarity?

Reviewer #1: No. Results presentation is unclear

Reviewer #2: The result section is well written

**Conclusions**

-Are the conclusions supported by the data presented?

-Are the limitations of analysis clearly described?

-Do the authors discuss how these data can be helpful to advance our understanding of the topic under study?

-Is public health relevance addressed?

Reviewer #1: (No Response)

Reviewer #2: The conclusion cannot be generalized to Sierra Leone, but only to the Masanga Hospital. Once the Title will be reconsidered, the conclusion will also be fine

**Editorial and Data Presentation Modifications?**

Reviewer #1: (No Response)

Reviewer #2: None

**Summary and General Comments**

Reviewer #1: Review comments

 ‘Chronic wounds in Sierra Leone: searching for Buruli ulcer, a NTD caused by Mycobacterium ulcerans’

General comments

The authors set out to identify BU among patients with ulcers that were suspected to be BU. While it is important to assess for presence of BU in Sierra Leone, there are several important issues that they need to address.

Major issues

Introduction:

Line 118: The authors state that surgical intervention is often required in BU. This is erroneous and needs to be corrected. Indeed, most cases of BU heal with only recommended combination antibiotic therapy (now oral clarithromycin and rifampicin) and appropriate wound dressing. At the present time, surgery is mainly adjunctive and may include procedures such as wound debridement and skin grafting. The treatment of BU including the role of surgery needs to be properly contextualised and discussed to avoid confusing non-experts who might read the manuscript.

Methods

The case definition for BU used is not clear. The authors state that ‘Suspicion of BU was based on several criteria, including: a lesion exhibiting one of the classical four forms of BU (painless itchy nodules, plaques, small ulcerations, extensive ulcerations)…Were non ulcers eg nodules and plaques included in the study?

If only ulcers were included, why do the authors say that fine needle aspirates were collected (line 138-139)? FNA is only recommended for non-ulcerative lesions. The authors need to explain this. Further from line 213-216, the authors indicate no FNAs were taken as no patients with non-ulcerative lesions presented. This section and the previous one referred to supra makes reading confusing and difficult to understand. The authors should rationalise the 2 sections to improve readability and understanding

Macrovascular complications of diabetes mellitus presenting with foot ulcers are an important cause of chronic lower limb ulcers in the West African sub-region. Did the authors not consider this aetiology? I find it difficult to appreciate why this important differential was not considered. Rather than focus on previous history of malaria and yellow fever, it will be more important to know which patients had history of diabetes and how diabetes mellitus was ruled out in others. 

While it is important to ascertain whether BU is present in Sierra Leone, it is not very helpful when the defined population for the study is not representative. In West Africa, BU typically occurs in persons <15 years and ulcers have been present for a relatively short duration. Why was BU being looked for in patients majority of whose ulcers had been present for longer than 1 year? Again, the age range of the population under study does not represent the typical BU population in West Africa. This is an important consideration as the likelihood of finding BU was low apriori 

In table 1, Participant 18 presents with a left foot ulcer which had been present for 36 months. Was this not a neuropathic ulcer related to his leprosy which he had as per table 2? How was this ruled out as a differential since patients with ‘other plausible cause of ulcer’ are listed as being excluded from the study?

Technique for collecting wound swabs for BU- why swab centre of the wound?

And why did the authors feel they will have to reconfirm (line 227) if qPCR, DSE or histopathology was positive for BU? The rationale for wanting this reconfirmation is unclear.

Although the authors say no FNA samples were taken as no patients with non-ulcerative lesions were included, they still describe DNA extraction and qPCR technique using FNA samples (line 249). Why?

Delete line 267-269 as it is of no added importance to the current manuscript.

Was disability assessed in these patients with chronic wounds? What were the findings of the disability assessment?

Was this a retrospective study? It seems to me a prospective study will have obtained necessary contact details of patients for follow up and communication. How did the authors intend to disseminate test results to patients if they returned positive and therefore needed appropriate antibiotic therapy for BU? 

What was the time interval between sample collection and when results were available from the European laboratory?

Did the participants receive any treatment for their wounds? How were the chronic ulcers managed? What wound care services were provided to participants? This has not been described except a brief mention in line 301 that ten lesions required surgical intervention during the study period. 

Results 

All 22 patients included in the study tested negative for BU on all modalities. Is this not an indication the awareness and expertise of the involved staff in recognising BU lesions is low? Some training may have been needed to improve the clinical recognition of BU especially in a setting as described where diagnostic facilities are limited. The authors should indicate practically how this can be achieved

What were the actual histopathology findings? What were they suggestive of?

The authors indicate a high rate of loss to follow up. Reading the methods section however, it is unclear if there was any real attempt or intent to follow up participants and what the intended purpose of any follow up was. Clarify 

Line 292-293: Repetition. Delete ‘limited to a single leg’

Line 301: ‘…ten had lesions requiring surgical intervention during the study period’. What was the reason(s) for the surgical intervention and what surgical procedures were performed?

Table 1: Needs to be redone. Why are ages and weights of individual participants presented as ranges? 

The authors state ‘Fifty percent of participants reported having previously undergone a surgical intervention, including split skin graft (36%), debridement (14%), and amputation (9%), although it was difficult to ascertain if this was at the same site as the presenting wound’. See Tables 1 and 2 for further details ’(line 313-316). There is no indication in the tables of which participants had prior surgical intervention. This must be included in the table 2. Again, although 50% were said to have had surgery, the total percentage of surgical interventions totals more than 50% [split skin graft (36%), debridement (14%), and amputation (9%)]. Why is this so?

The authors also say ‘Fifty percent of participants reported having previously undergone a surgical intervention, including split skin graft (36%), debridement (14%), and amputation (9%), although it was difficult to ascertain if this was at the same site as the presenting wound’. Why is it difficult to ascertain the site of surgical interventions like grafting or an amputation? This sounds a bit cryptic to me and some logical explanation is required as it seems to my mind that it should be possible to ascertain a grafting or amputation site.

What was the interval between the reported surgical intervention and the time of sampling for BU confirmation? This is an important consideration that can impact results of laboratory confirmation of BU

It is not clear what the authors seek to achieve by providing previous histories of malaria and yellow fever. Is there a relationship with the current wounds?

In Table 2, several persons are reported to have weight loss; in association with chronic ulceration, is there a suspicion of the ulcers being malignant? There is a need to explain the possible reasons for the weight loss. Again, under weight loss column, some are described as ‘incomplete’. What does that mean? 

Participant 17 has ‘incomplete’ palpable lymph nodes. Meaning?

Line 328-329: Which specific cocci and rods were found to be present in the wounds? This is important to indicate

Line 371. Change to ‘a well-known risk factor…

(Line 379-380): How does this study ‘address a knowledge gap in the understanding of the aetiology of chronic lower limb ulceration in Sierra Leone’? 

Line 394-396: ‘In addition, no participants presented with nodules or plaques, which may be because these are typically painless and therefore without an impetus to seek treatment if knowledge of BU is lacking in the community’. This is not a reasonable conclusion from this study as the initial study population comprised patients presenting with only wounds/ ulcers in the hospital. Please delete

Reviewer #2: This paper described the result of the search of Buruli ulcer in an Hospital of Sierra Leone. It resulted that none of the lesions examined were not consistent with BU. The Authors, thus, wrightly, concluded that, this negative result did'nt discard existance of BU in Sierra Leone. Indeed, the relevance of this study, given the fact that participants are not recruited from a community case search study, and the very limited number of participants, are valid only for this study site, namely the Masanga Hospital. We, then, suggest to the authors, to review the scope of the study, reformulating the title, to limit it to the hospital site

PLOS authors have the option to publish the peer review history of their article (what does this mean?). If published, this will include your full peer review and any attached files.

Reviewer #1: No

Reviewer #2: Yes: Ghislain E. Sopoh
---

## [Editor Report · Decision Letter 1]

10 Aug 2021

Dear Miss Please,

Thank you very much for submitting your manuscript "Chronic wounds in Sierra Leone: searching for Buruli ulcer, a NTD caused by Mycobacterium ulcerans, at Masanga Hospital." for consideration at PLOS Neglected Tropical Diseases. As with all papers reviewed by the journal, your manuscript was reviewed by members of the editorial board and by several independent reviewers. The reviewers appreciated the attention to an important topic. Based on the reviews, we are likely to accept this manuscript for publication, providing that you modify the manuscript according to the review recommendations. 

Thank you for your revision of your manuscript.

I have a few suggestions to follow up on the recommendations of the original reviewers.

1. Short Title: Searching for Buruli ulcer at Masanga Hospital, Sierra Leone

2. Author Summary, lines 106-107: These results suggest that BU is not a major cause of chronic wounds in patients presenting at Masanga Hospital in central Sierra Leone. (When I look at the map, Masanga appears to be in the central part of the country. On line 155, you state that it is in the north of Sierra Leone.)

3. line 197: change semi-colon to comma

4. line 143: anti-mycobacterial (e.g., clarithromycin is not typically used in the treatment of tuberculosis).

5. In the introduction, you state that only 28 cases were reported in 2011 following a single case in 2008. Would it be possible to ascertain where in Sierra Leone these cases were found either from The WHO, the Ministry of Health, or other resource. BU is typically focal and often transient. For example, in Benin, it's in the south but in Ghana, cases are often further inland. You should address this issue in the Discussion.

Sincerely,

Paul J. Converse

Associate Editor

Mathieu Picardeau

Deputy Editor

Dear Miss Please,

Thank you for your revision of your manuscript.

I have a few suggestions to follow up on the recommendations of the original reviewers.

1. Short Title: Searching for Buruli ulcer at Masanga Hospital, Sierra Leone

2. Author Summary, lines 106-107: These results suggest that BU is not a major cause of chronic wounds in patients presenting at Masanga Hospital in central Sierra Leone. (When I look at the map, Masanga appears to be in the central part of the country. On line 155, you state that it is in the north of Sierra Leone.)

3. line 197: change semi-colon to comma

4. line 143: anti-mycobacterial (e.g., clarithromycin is not typically used in the treatment of tuberculosis).

5. In the introduction, you state that only 28 cases were reported in 2011 following a single case in 2008. Would it be possible to ascertain where in Sierra Leone these cases were found either from The WHO, the Ministry of Health, or other resource. BU is typically focal and often transient. For example, in Benin, it's in the south but in Ghana, cases are often further inland. You should address this issue in the Discussion.

Figure Files:

Data Requirements:

Reproducibility:

References

---

## [Editor Report · Decision Letter 2]

28 Sep 2021

Dear Miss Please,

We are pleased to inform you that your manuscript 'Chronic wounds in Sierra Leone: searching for Buruli ulcer, a NTD caused by Mycobacterium ulcerans, at Masanga Hospital.' has been provisionally accepted for publication in PLOS Neglected Tropical Diseases.

Best regards,

Paul J. Converse

Associate Editor

Mathieu Picardeau

Deputy Editor

---

## [Editor Report · Acceptance letter]

8 Oct 2021

Dear Miss Please,

We are delighted to inform you that your manuscript, "Chronic wounds in Sierra Leone: searching for Buruli ulcer, a NTD caused by *Mycobacterium ulcerans*, at Masanga Hospital.," has been formally accepted for publication in PLOS Neglected Tropical Diseases.

Best regards,

Shaden Kamhawi

co-Editor-in-Chief

Paul Brindley

co-Editor-in-Chief
